# The Effects of Sequencing Strategies in Teaching Methods on Nursing Students’ Knowledge Acquisition and Knowledge Retention

**DOI:** 10.3390/healthcare10030430

**Published:** 2022-02-25

**Authors:** Wei-Ting Lin, Ching-Yun Yu, Fan-Hao Chou, Shu-Yuan Lin, Bih-O. Lee

**Affiliations:** 1College of Nursing, Kaohsiung Medical University, Kaohsiung 80708, Taiwan; waittea@gmail.com (W.-T.L.); cyyu@kmu.edu.tw (C.-Y.Y.); biholee@kmu.edu.tw (B.-O.L.); 2Department of Medical Research, Kaohsiung Medical University Hospital, Kaohsiung 80708, Taiwan; 3Center for Medical Education and Humanizing Health, Kaohsiung Medical University, Kaohsiung 80708, Taiwan; fanhao@kmu.edu.tw

**Keywords:** problem-based learning, sequence of teaching, knowledge acquisition, knowledge retention

## Abstract

*Background*: No existing research has determined which teaching sequence strategy is the best for nursing students. *Purpose*: To find out which sequence is most effective in knowledge acquisition and knowledge retention and to further verify knowledge acquisition between problem-based learning (PBL) and lecture-based learning (LBL). *Methods*: This was a quasi-experimental design with a comparison of two nursing student groups selected from students in their final program year who were invited to participate. Generalized estimating equation was used to compare Group I (LBL-PBL-clinical practicum) and Group II (PBL-LBL-clinical practicum) by using knowledge acquisition and knowledge retention as outcome variables. *Findings*: Fifty-six senior students joined this study. Group I was significantly better than Group II on both knowledge acquisition (β = 7.05, *p* = 0.04) and knowledge retention (β = 9.40, *p* = 0.03). *Discussion*: The sequence of LBL-PBL-clinical practicum or policy of allowing practicum and courses in the same semester might be the best strategy to enhance knowledge retention.

## 1. Introduction

Nursing education needs to change to meet the expectations of employing hospitals and clinics that new graduates demonstrate not only performance of good clinical skills but also inter/intrapersonal skills, such as communication, teamwork, critical thinking, problem-solving, and so on [1]. Problem-based learning (PBL) is considered one of the effective strategies that may fit hospital demands; additionally, it promotes students’ communication, teamwork, problem-solving, self-direct learning, critical thinking, and respect for others [2,3].

Combining the benefits of both PBL and lecture-based learning (LBL) might improve the effectiveness of learning outcomes in meeting the requirements of employers. Although LBL is a traditional teaching method in nursing education that might not be preferred by students, LBL provides a structure with the sequence of each subject and functions as a standard textbook. The aim of LBL is to identify the key knowledge of each subject to all students [4], whereas problem-based learning (PBL) motivates students to identify their own structure of knowledge toward learning in each subject.

In The Wiley Handbook of Problem-Based Learning, Moallem pointed out that the effect of PBL in terms of students’ academic performance has yielded inconclusive results [5]. Although the majority of studies have compared the effectiveness between PBL and LBL, most have found that PBL is significantly better for students’ academic performance [6,7], although a few found no significant difference in students’ academic performance [8]. Sá, Amarante et al. studied the topic of teaching sequence rather than different teaching methodologies in a biology course and concluded there were no differences in the sequence of class design [9]; consequently, there appears to be no definitive answer from the literature. A meta-analysis by Sayyah et al. indicated that PBL was more favorable than a combined integration of PBL and LBL [10]; however, Yue et al. indicated that integrating PBL and LBL produced significantly better academic performance than PBL alone [11].

In Taiwan, the nursing administration clinical practicum has been reduced from a three-credit course to a two-credit one by university policy. The nursing administration project was designed as one topic of a two-credit course curriculum, and this became a key concept to apply in the nursing administration clinical practicum. The nursing administration project uses quality improvement theory and practice in a clinical setting, which is similar to the process of PBL: problem analysis, self-directed individual learning, and a reporting phase [12,13]. The first step of the nursing administration project is to find and analyze the problem in clinical settings. Secondly, with self-direction, students learn to find what has been reported in the literature or in similar clinical settings and then generate solutions. Thirdly, students report this in their on-course homework; therefore, PBL is a perfect tool to train students prior to clinical practicum in the nursing administration project. This study uses Kolb’s concepts of the experiential learning model to illustrate the functions of PBL, LBL, and clinical practicum and how sequences of teaching methods impact the learning process [14].

## 2. Background

Although PBL has been used in education for decades, Taiwan has promoted this teaching method only in the last 10 years. The Ministry of Education implemented the Higher Education Teaching Excellence Project that promoted PBL in higher education in 2005. PBL was implemented in mostly medical-related programs in universities. For example, medical programs were started earlier than other programs [15]. Nursing [16], pharmacy [17], and psychology programs [6] are the other medical-related programs that have used PBL.

In Taiwan, courses usually combine PBL and LBL in the same topic, while other studies have shown that each topic uses either PBL or LBL [2,15]. Studies have only focused on comparing test scores right after PBL or LBL program completion [6] or have looked specifically at self-directed learning [3], critical thinking [18], or self-efficacy [19]. Nursing education is a practical science in which students need to make the connections between theory and practical skills and then apply these to real-time situations. Knowledge is built up by an accumulated learning process. In addition, teaching methods such as PBL and LBL have been used simultaneously at different times of the academic year. To evaluate learning results after clinical practicum is the true concern of nursing education. Although both PBL and LBL can help students transfer learning knowledge to clinical practice, determining which teaching method to use first might help students learn better is regarded as being more important.

This study applied the experiential learning model [14] to demonstrate the facilitation of PBL, LBL, and clinical practicum in the stages of learning and two teaching sequences leading to different learning outcomes (Figure 1). The four-stage cycle of learning in Kolb’s model was applied in this study [14]. Kolb’s theory proposes that experiential learning theory is an adult knowledge development model that emphasizes experiences as the key element in a learning process. The theory postulates that knowledge is created through the transformation of the learning experience and built through engaging in all phases of a learning cycle [14]. The cycle usually starts with a concrete experience. In the first stage, students need to be open-minded and embrace new experiences without judgment. This is followed by the second stage of observation and reflection; the students observe these experiences with different perspectives and reflect on their own experiences. The third stage is abstract conceptualization, in which the students conclude an abstract of concepts by incorporating observations, reflections of experiences, and the generation of logical concepts. The final stage is active experimentation, in which the students activate the logical concepts and test these in an appropriate situation.

The process of PBL has three phases: problem analysis, self-directed individual learning, and a subsequent reporting phase [13,20] (Figure 1). In PBL settings, teachers usually give students problems that need to be solved. The students start the first phase, problem analysis, which triggers the learning cycle [13]. While analyzing the problems, some doubt or confusion occurs that activates students to collect prior knowledge and find resources. These situations engage students in learning and lead into the phase of self-directed individual learning [13]. Finally, in the subsequent reporting phase, the students have to report what they have learned from the self-directed learning to the group [13]. The students engage with their peers’ learning through collecting each other’s prior knowledge, discussing each other’s findings, and consolidating the concepts of what students have learned and reported back to the group [21]. The students then move on to new problems and restart the cycle.

One study proposed that the lecture could be a facilitator to the first step of learning, which provides a very basic introduction to the learning experience (concrete experience) in Kolb’s [14] model. PBL aims to involve students in active learning by emphasizing the experience of the learning cycle [22]. The problem analysis and self-directed individual learning phase of PBL could facilitate students to access their prior knowledge and reflect on their previous learning (reflective observation). In the subsequent reporting phase, students have to summarize what they have learned from self-directed or group learning that then enhances the formation of new concepts (abstract conceptualization). A clinical practicum may boost students to actively apply what they have learned from the course (active experimentation).

PBL might not be effective in knowledge acquisition. The majority of the studies concerning knowledge acquisition make comparisons between PBL and LBL. Some studies have found PBL is significantly better in knowledge acquisition than LBL [6,23], although others found no significant difference in knowledge acquisition [8,24]. Kazi et al. even reported that students who learned via PBL had less knowledge acquisition compared with students learning via LBL [24].

Sangestani et al. found that a group with LBL performed significantly better than a group with both LBL and PBL methods among midwifery students in pregnancy and childbirth courses [25]. Pourshanazari et al. found that a PBL group had a higher score than an LBL group on respiratory physiology skills but not tests among medical students [26]. Wijnen et al. studied first-year psychology university students and compared a PBL to an LBL group and found that the PBL group was better on both immediate and delayed tests [6].

Khoshnevisasl et al. studied a pediatric course of medical students and used randomized control trials to compare PBL with LBL, and the result indicated that PBL was better than LBL, but this did not reach a significance level [8]. However, these three studies did not look at how students actually learned after the clinical practicum. These studies might not have truly investigated or understood the teaching methods that influence knowledge acquisition in students that have a clinical practicum.

Meta-analyses have also yielded different results. In an earlier meta-analysis, PBL was not significantly better in knowledge acquisition, but in a later meta-analysis, Zhang et al. found better yields of knowledge acquisition in PBL than LBL, but with a small effect size due to low-quality studies [23]. Schmidt et al. found that students in PBL gained slightly less knowledge, especially in basic science subjects, than did students in LBL because students in PBL were expected to focus on applying theory and solving clinical problems [27]. On the contrary, Zhang et al. showed that in a PBL group course examination, general pass rates and high excellence pass rates were significantly better than those for the LBL group [23]; however, around half of the studies that were included in the meta-analysis exhibited low-quality scores. Zhang et al. further concluded that PBL was more effective when applied to laboratory courses than to theory-based courses [23]. Most studies in the literature used test scores after interventions, for example, PBL, TBL, simulation, and team-based learning, to measure knowledge-related issues. Most of the studies measured group scores within one week after intervention [6,28]. Studies used multiple-choice [4,29] or both choice and non-choice questions [7], but most studies used self-developed tests according to the study area [7,29], with only a few using standardized tests from creditable institutions; for example, the American Heart Association, Health Education System, Inc., and The Pharmacist and Patient-Centered Diabetes Care Certificate Training program of American Public Health Associations [17]. Some self-developed tests were verified by the expert of the area [7,29], while others did not specify the validity of the test [6].

PBL may be effective for knowledge retention. Some unverified thinking assumes that learning strategies can automatically result in knowledge retention. Zieber et al. implemented their study in undergraduate nursing students by questioning this assumption and proposed that knowledge retention needs to be designed in action and by activity [29]. PBL focuses on developing students’ rich and flexible knowledge, providing opportunities for students to review prior knowledge, and applying such learned knowledge in problem analysis, acquiring new knowledge, and reporting to group members. Therefore, PBL is a good activity to enhance knowledge retention. The questions as to whether PBL is more effective than LBL in terms of knowledge retention have yielded inconclusive results [5]. Wijnen et al. found no differences in knowledge retention between LBL and PBL groups because the decline in performance over time was equal in all conditions [6]. On the other hand, some researchers have found that for longer-term knowledge retention, PBL has superior efficacy over LBL [13,26].

Schmidt et al. determined that PBL had an accumulated learning effect [27], and, especially for students who had been exposed to PBL from junior classes, knowledge would grow over their clerkship. Yew et al. further concluded that PBL was strongly influenced by earlier phases and would increase students’ self-directed learning ability, thus increasing knowledge retention [12].

PBL provides students with not only emotional stimulus through group discussion but also effective motivation through group collaboration, thus possibly improving knowledge retention. Chittaro et al. stated that negative emotional arousal was a factor to increase retention [30]; specifically, memory retention was related to emotional intensity and was aroused by an experience. Levy further pointed out the three components that compose the process of knowledge retention: defining the desire for preservation of knowledge, documenting the knowledge, and, finally, integrating it into daily life [31].

Similar to knowledge acquisition, most of the studies used tests to measure knowledge retention. However, when to test students is the key to measuring such retention. Zieber et al. used a test immediately after the program, and Wijnen et al. used a test one week after intervention [6,29]. Bowers et al. used one to three months later, while Doomernik et al. used one 17 months later to measure knowledge retention [17,32].

## 3. Methods

### 3.1. Aims

The main aim of this study was to find out which sequence of teaching strategies was more effective for students in both knowledge acquisition and retention and to further verify the effects between PBL and LBL.

### 3.2. Design

This was a quasi-experimental study with the comparison of two nursing student groups using repeated measures.

### 3.3. Settings and Participants

The nursing administration course is a two-credits-required course with a further two credits clinical practicum in the fourth (last) year of the nursing university. The student population total is 68 in the study year. The study only focused on one topic: the nursing administration project, which was designed as two hours of LBL, three sections of two hours of PBL, and 10 days of clinical practicum.

The sample was recruited from the nursing administration course who were enrolled in their final year at the university by free will. By university rules, students take the nursing administration course in the fall semester and the clinical practicum in the fall or spring semester. Group I had the clinical practicum in the fall semester, and Group II the same in the spring semester. Students were selected based on their situation such as accessibility of vehicles and overseas student exchange program.

There were 56 students who joined this study for an academic year, and no student was excluded. The sample was 82% of the main population. Students were divided into eight groups, with 8–10 students per group. The main instructor of the course taught both LBL and PBL modalities. Seven other instructors helped instruct the PBL sessions.

G*Power software (version 3.1.9.2) was used to estimate the required sample size. To estimate the comparison of four time points of repeated measures with a median effect size of 0.25, an alpha = 0.05, and a power = 0.8, this study required at least 48 students. As such, the 68 students who participated should be sufficient.

### 3.4. The Intervention

The cycle of the learning process of the two groups is shown in Figure 1. To design the different sequence of teaching methods in this study, Kolb’s experimental learning model was adopted [14]. The model could demonstrate the learning phase, and the process of PBL learning could also fit into the experiential learning model, which could explain the effect sequence of teaching methods on knowledge acquisition and retention of the students. Group I was exposed to the sequence of LBL-PBL-clinical practicum, while Group II was exposed to the sequence of PBL-LBL-clinical practicum. The learning cycle of Group II was disrupted from the phase of reflection observation and returned to phase one, concrete experience, which should have led to reflective observation. However, the cycle was pushed forward directly to the active experimentation phase. The disruption of the learning cycle might have led to fragmented knowledge abstraction, and such a fragile foundation of the knowledge might possibly have led to less knowledge retention.

### 3.5. Data Collection

The two groups both took the pretest at the beginning of the fall semester (Time 1, T1) from September to January. Group I had a lecture-based class first with class test (Group I Time 2, G1T2), followed by PBL sessions with a test (Group I Time 3, G1T3), and with a clinical practicum in the fall semester. Group II had PBL sessions first with test (Group II Time 2, GIIT2) and had a lecture-based class at the beginning of the spring semester with a test (Group II Time 3, GIIT3) followed by a clinical practicum. All the groups had another follow-up test around one month after the clinical practicum (Time 4, T4). The specific times are shown in Table 1. Intervention and data collection were scheduled over the academic year.

### 3.6. Ethical Consideration

The study protocol was approved by the Institute Review Board (IRB) at the university hospital (IRB NO XXXX). The instructor explained the purpose of the study and assured the subjects that participation was voluntary and that they had the right to withdraw at any time without any penalty. Students willing to participate were requested to sign the written consent form held by the teaching assistant.

### 3.7. Data Analyses

IBM SPSS 20.0 was used to analyze research data. T-tests were used to analyze the differences between Groups I and II. Generalized estimating equation (GEE) was used to accommodate the variation in correlation between repeated measures and the test scores as an outcome variable. The test scores of Time 1 had a significant difference; therefore, the statistical method was adjusted by using the test scores of Time 1 as a control variable to account for the differences between Groups I and II. An autoregressive (AR(1)) correlation structure was used to accommodate the repeated measures with higher correlation across time [33].

### 3.8. Validity and Reliability of the Instruments

The knowledge test followed the guidelines of the nursing administration project provided by the Taiwan Nurses Association. Knowledge tests used exactly the same items four times. The test consisted of three parts that totaled 100 points that were composed of eight multiple-choice questions (40 points), one fill-in-the-blank question (5 points), and four open-ended questions (55 points). Academic experts in the nursing administration project examined the validity of the test questions. The primary instructor alone graded the open-ended questions to ensure grading consistency and inter-rater reliability. The whole project was approved by the Lecture Board of the College of Nursing at the study university.

Knowledge acquisition was measured within one week after intervention, which is similar to studies as reported in the literature review. The knowledge assessments were made four times. Time 1 measured the knowledge base before student learning about the nurse administration project, Time 2 measured knowledge acquisition of LBL or PBL, Time 3 measured knowledge acquisition within one week after students finished the LBL-PBL-clinical practicum or PBL-LBL-clinical practicum, while Time 4 measured knowledge retention in the sequence of the LBL-PBL-clinical practicum or the PBL-LBL-clinical practicum. Knowledge retention was measured around one month after the clinical practicum.

### 3.9. Findings

Fifty-eight senior students were enrolled in this study, with the mean age of 28 students in Group I being 23 ± 0.68 and the mean age of 28 students in Group II being 23 ± 0.63. There was no significant difference between groups (t (54) = 1.50, *p* = *0*.14). The mean differences between Group I (LBL-PBL-clinical practicum) and Group II’s (PBL-LBL-clinical practicum) intervention are shown in Table 2.

Group I (LBL-PBL-clinical practicum) and Group II (PBL-LBL-clinical practicum) had an intersection after Time 3 (see Figure 2). Figure 2 indicates that the test scores of Group I gradually improved and became even better than Group II at Time 4. The result of the GEE indicated that there was a significant difference in Time 1 (β = 0.46, *p* = 0.00), and between groups (β = −4.23, *p* = 0.01). Comparing three other times to Time 1 yielded different results (β = 10.28–5.58, *p* = 0.00–0.09).

There was no significant difference between groups when comparing Time 2 to Time 1 (β = 2.35, *p* = 0.56). There was no significance in students’ score differences between PBL and LBL groups; however, at Time 3, Group I was significantly better than Group II (β = 7.05, *p* = 0.04). Comparing Time 4 to Time 1, Group I had significantly higher scores than did Group II (β = 9.40, *p* = 0.03) (Table 3). Therefore, knowledge retention was better in the sequence of the LBL-PBL-clinical practicum than in that of the PBL-LBL-clinical practicum.

## 4. Discussion and Recommendation

The first main finding of this study is that knowledge acquisition was significantly better using the sequence of the LBL-PBL-clinical practicum than that of the PBL-LBL-clinical practicum. From Kolb’s learning cycle, once students have reached abstract conceptualization, the disruption and reversal of concrete experience (first step) might disrupt the process of learning [14]. The reversal causes fragmented knowledge abstraction and a more fragile foundation of knowledge, possibly leading to less knowledge acquisition. The process of PBL emphasizes that students have to access their prior knowledge to answer the questions; thus, a stronger foundation from lectures will allow students to develop stronger knowledge through a clinical practicum.

To the best of our knowledge, this was the first study to look at the sequence of different teaching methods after clinical practicum that influences knowledge acquisition and its effects, although there was no confirmatory evidence from other studies. Some studies looked at the academic achievement after clinical practicum but not specifically at the effects of PBL on knowledge acquisition. Sangestani et al. compared a group with LBL and groups with both LBL and PBL, although the methodology might not have revealed whether the teaching method works or whether the amount of exposure to the same material had actually illustrated the effect of the sequence of the teaching method [25]. Wijnen et al. found that knowledge acquisition was not significantly better than the group with LBL after controlling for the pretest [6]. Zhang et al.’s study concluded PBL was more effective in applying science- rather than theory-based courses by extending the effect of PBL; however, this study further emphasized that the sequence of an LBL-PBL-clinical practicum could maximize the effect of PBL [23].

The sub-finding of this study was that knowledge acquisition between LBL and PBL does not exhibit a significant difference. This sub-finding was similar to an early meta-analysis that found no significant difference between PBL and LBL exists, but PBL is slightly better than LBL [27]. However, this study did show similarity with Zhang et al.’s meta-analysis, which showed that the PBL group had better performance in passing examinations, but this result did not reach significance [23].

The second main finding was that knowledge retention was significantly better in the group with the sequence of the LBL-PBL-clinical practicum rather than the PBL-LBL-clinical practicum. This study looked at the effect after clinical practicum using different sequences of teaching methods in knowledge retention. Similar to other studies of knowledge acquisition, such studies also compared knowledge retention between PBL and LBL, but the results were inconclusive. However, the previous research might have missed one point that the ultimate goal of nursing education is to look at how nursing students perform in clinical settings.

For example, Ibrahim et al. merely looked at knowledge retention at certain points, not after clinical practicum [34]. Zieber et al. proposed that design activity may increase knowledge retention [29]. This study used LBL, PBL, and clinical practicum, which all help develop students’ rich and flexible knowledge and provide further opportunities for students to review prior knowledge and apply such knowledge in real-time situations. Schmidt et al. concluded that PBL has an accumulated learning effect [27]. Therefore, this study design could test such accumulated learning effect, although in this study, both Groups I (LBL-PBL-clinical practicum) and II (PBL-LBL-clinical practicum) retained more knowledge compared with each previous knowledge test (T4 > T3 > T2 > T1). However, it is harder to explain whether the accumulated learning effect could only be sourced from PBL. The current study is also contradictory to the finding of Wijnen that knowledge retention in both LBL and PBL had declined [6]. From the perspective of Kolb’s experimental learning cycle, the stronger the knowledge that is built, the stronger the retention [14]. Group I (LBL-PBL-clinical practicum) had both a clinical practicum and PBL in the same semester. This also implies that to maximize learning outcomes, schools should establish policies to allow students to complete the course and clinical practicum during the same semester.

Stress can be considered as emotional arousal, thus increasing knowledge retention [30]. On the other hand, students concentrated on the nursing administration project more, as they would use it immediately. This corresponds to Levy’s first components of the knowledge retention process, defining the desire to preserve knowledge [31]. The second step was students having to document such knowledge occurring in the process of PBL that they have to present to classmates. The third step of integrating all this into daily life was used in the application of the clinical practicum. The major difference between Group I (LBL-PBL-clinical practicum) and II (PBL-LBL-clinical practicum) is the first step.

By regular the academic design of the curriculum, Groups I and II both had courses in the fall semester, and Group I had a clinical practicum in the fall semester and Group II in the spring semester. A portion of students in Group I had PBL and a clinical practicum simultaneously. According to university policy, it is impossible to finish all nursing administration courses and PBL before a clinical practicum; therefore, the simple effect of PBL in Group I was not possible; rather, a combination of PBL with clinical practicum was more feasible. Additionally, Group I (PBL-LBL-clinical practicum) concentrated on this for over half the academic year, but Group II (LBL-PBL-clinical practicum) had the whole academic year. It could be argued that the sequence of these two designs was not parallel. Future studies should divide both Group I and Group II into LBL-PBL and PBL-LBL, which could reduce the unparalleled issue. The effect of PBL that provides student-centered learning is in no doubt. The competency of the instructor might also be a determining factor to facilitate the student rather than the technique [35]. Student preferences, learning style, or personality might be other important factors that influence the outcome of student learning. The instructor further influences the success of PBL because they need to be the facilitator across not only the PBL session but also the full program [36]. However, this study did not actually account for differences among PBL instructors, so further research is needed.

Based on the findings from this study, not only the short-term effects of PBL but also how LBL and PBL influence clinical performance and result in knowledge acquisition and knowledge retention could be determined. Since previous research has only looked at simply PBL and LBL, rather than the sequence, as strategies, whether placing LBL into Group II (PBL-LBL-clinical practicum) would enhance students’ clinical practicum needs further examination. It can also be applied to other nursing courses to allow the different selection of teaching methods, thus promoting nursing students’ learning outcomes.

## 5. Limitation

Limitations to the methodology design are Group I completing the study in a total of five months, while Group II completed the study over an eight-month period. As per the study university regulations, course design and clinical practicum could not be coalesced into one semester for both groups; therefore, using only one university as the sample limits generalization to other settings.

## 6. Conclusions

To the best of our knowledge, this study was the first to study the effect of the sequence of different teaching methods, including LBL, PBL, and a clinical practicum, on knowledge acquisition and retention. LBL was not more effective than PBL in knowledge acquisition. However, the sequence of the LBL-PBL-clinical practicum was better than the PBL-LBL-clinical practicum sequence on knowledge acquisition and retention. Corresponding to the study result, the course and clinical practicum should ideally be in the same semester, and further study is needed in the area of whether LBL is needed after PBL. The sequencing of teaching methods might also be applied to other nursing courses to enhance the learning outcome and promote their professional competency in improving quality of care. The sequencing of teaching methods might also indicate that a policy allowing a simultaneously run course and clinical practicum could enhance student learning outcomes. Future studies are also needed, including testing the model in different nursing courses for such validation.

## Figures and Tables

**Figure 1 healthcare-10-00430-f001:**
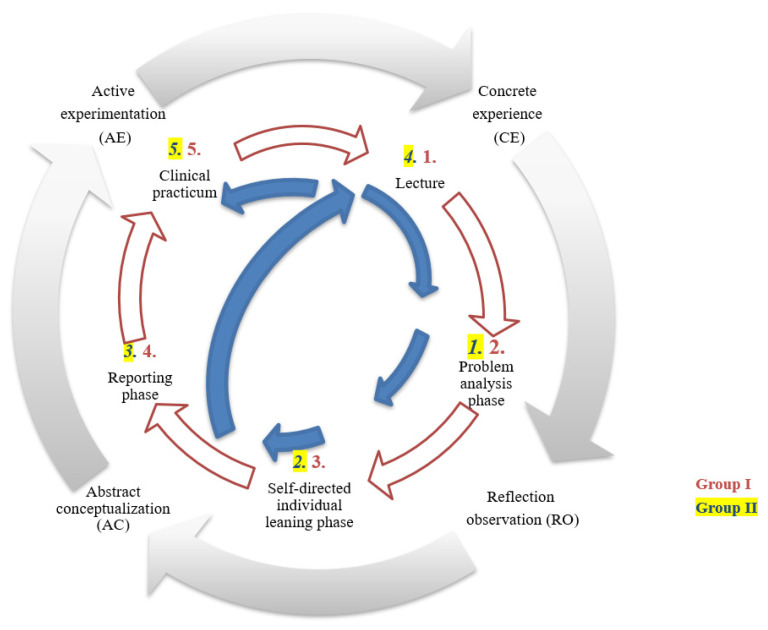
The different phases of problem learning, lecture-based learning, and clinical practicum in the experimental learning model between Group I and Group II.

**Figure 2 healthcare-10-00430-f002:**
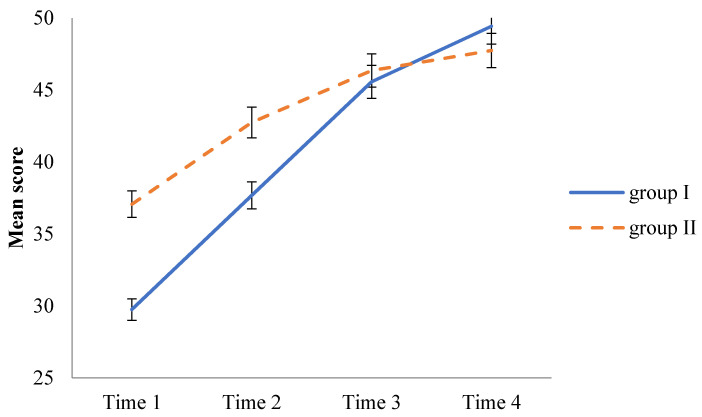
Patterns of scores through four time schedules between Groups I and II.

**Table 1 healthcare-10-00430-t001:** Intervention and data collection schedule in an academic year.

	SEP	OCT	NOV	DEC	JAN	FEB	MAR	APR	MAY
Test	GIT1	GIT2		GIT3	GIT4				
	GIIT1			GIIT2		GIIT3			GIIT4
PBL		GI,II	GI,II	G1,II					
Clinical practicum			GI	GI			GII	GII	
Lecture		GI				GII			
Note. T1, Time 1; T2, Time 2, T3, Time 3; T4, Time 4; GI, Group I; GII, Group II.									

**Table 2 healthcare-10-00430-t002:** The mean scores across four time schedules between Group I and Group II’s interventions.

	Group I (*n* = 28)	Group II (*n* = 28)	t	*p*
Time 1 Intervention	Pretest	Pretest		
Time 1 score	29.75 ± 8.93	37.07 ± 9.23	−3.02	0.00 *
Time 2 Intervention	Lecture	PBL		
Time 2 score	37.68 ± 12.69	42.75 ± 14.72	−1.38	0.17
Time 3 Intervention	PBL	Lecture		
Time 3	45.57 ± 11.59	46.36 ± 10.66	−6.75	0.79
Time 4 Intervention	Clinical Practicum	Clinical Practicum		
Time 4	49.43 ± 15.88	47.75 ± 13.72	0.42	0.67

*p* < 0.05 *.

**Table 3 healthcare-10-00430-t003:** Patterns of relationship among times and groups in test scores.

Parameter	β	Std. Error	95% Wald Confidence Interval		Hypothesis Test		
			Lower	Upper	Wald Chi-Square	df	Sig.
Intercept	20.20	4.19	11.98	28.42	23.19	1.00	0.00 *
Time1	0.46	0.11	0.24	0.68	17.13	1.00	0.00 *
Group I vs. II	−4.23	1.64	−7.43	−1.02	6.68	1.00	0.01 *
Time 4 vs. Time 1	10.28	3.37	3.67	16.89	9.29	1.00	0.00 *
Time 3 vs. Time 1	8.77	2.73	3.42	14.13	10.30	1.00	0.00 *
Time 2 vs. Time 1	5.58	3.25	−0.78	11.94	2.96	1.00	0.09
Time 4 vs. Time 1 * Group I vs. II	9.40	4.32	0.93	17.86	4.74	1.00	0.03 *
Time 3 vs. Time 1 * Group I vs. II	7.05	3.46	0.26	13.83	4.14	1.00	0.04 *
Time 2 vs. Time 1 * Group I vs. II	2.35	4.04	−5.57	10.27	0.34	1.00	0.56

*p* < 0.05 *.

## Data Availability

The data are not publicly available due to their containing information that could compromise the privacy of research participants.

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
