# Peer review of "The Effects of Sequencing Strategies in Teaching Methods on Nursing Students’ Knowledge Acquisition and Knowledge Retention"

_healthcare, 2022, doi:10.3390/healthcare10030430_

Round 1

Reviewer 1 Report

#1.  Introduction, Background

This study did a good job specifically explaining the critical evaluation of the scientific literature, the gaps among the literature, and the need in the educational environment. I recommend adding a detailed explanation of Kolb’s concepts of the experiential learning model to this study.

#2. Settings and Participants

1) I recommend that you describe the sampling method of your study.

2) In research, the criteria for exclusion of subjects are important. Please describe the criteria for exclusion of subjects.

#3. Validity and reliability the instruments

Please describe whether the tool is approved for use and its reliability.

Author Response

Thank you so much for the efficient review process and valuable feedback from the reviewers. We are submitting a revised manuscript entitled “The Effects of Sequencing Strategies in Teaching Methods on Nursing Students’ Knowledge Acquisition and Knowledge Retention” to the journal of Healthcare for the consideration of publish. To respond the reviewer’s comments and suggestions, we list the questions point by point and the revisions we have made on the manuscript as follows.

With Best Regards,

Shu-Yuan Lin, PhD, RN. Professor,
Director of School of Nursing, Kaohsiung Medical University, Taiwan
Email: [email protected]

Reviewer 2 Report

  • The manuscript is well-written and highlights an important and emerging teching method PBL Vs LBL in nursing education research area.

    There are few things that should be revised or included by the authors:

  • - Measuring the quality of knowledge-related research tools There were no reports in the quality assessment of research tools such as measurements, tests, exams that assess the sample's knowledge.

- Appropriate sample size the researchers reported How, how much, and what references have been used to determine the size of the group, and what references are there to support the academics?

- PBL vs LBL, what's there to support different time periods? of the lessons that have taken place What's the point of discussion? to support, please clarify.

- Inclesion Criteris, what are the criteria for selection into Group 1 and Group 2? Please clarify what characteristics should be similar for both groups.

- At least the sample went or not? should be described as a limitation of education or suggestions to increase the number or not, please specify

- The results of the study found that the two teaching methods were not different. Therefore, what recommendations are important in this research? To publish outstanding to be reported to readers to be informed. and guidelines for utilizing the research, please specify and add. Reply to reviewer.

Author Response

(The authors gave the same response as above.)
